# Anti-Apoptotic c-FLIP Reduces the Anti-Tumour Activity of Chimeric Antigen Receptor T Cells

**DOI:** 10.3390/cancers14194854

**Published:** 2022-10-04

**Authors:** Grace Min Yi Tan, Aarati Poudel, Seyed Mohammad Ali Hosseini Rad, Alexander Donald McLellan

**Affiliations:** 1Department of Microbiology and Immunology, University of Otago, Dunedin 9010, New Zealand; 2Department of Molecular and Clinical Cancer Medicine, University of Liverpool, Liverpool L1 1NT, UK; 3Kite Pharma, 1800 Stewart St., Santa Monica, CA 90404, USA

**Keywords:** CAR T cells, solid tumours, AICD, anti-apoptotic, c-FLIP

## Abstract

**Simple Summary:**

Chimeric antigen receptor (CAR) T cell treatment is a promising adoptive cell therapy that utilises CAR-expressing primary T cells to target specific tumour antigens. Since its first approval in 2017, the FDA has approved six CAR T cell therapies for blood cancer treatment. Despite their success in treating B cell malignancies, solid-tumour-directed CAR T cells face multifactorial challenges, often resulting in a lack of anti-tumour response in tumour elimination. The aim of this study was to evaluate the potency of expressing anti-apoptotic gene FLICE-like inhibitory protein p43 (c-FLIPp43) in Her2-CAR T cells to resist activation-induced cell death (AICD). This study confirmed the anti-apoptotic activity of c-FLIPp43. However, the expression of c-FLIPp43 reduced the CAR activity in a Her-2 breast cancer xenograft model relative to the control Her2-CAR T cells. This work provides insight into the role of c-FLIP in T cells and its impact on anti-tumour immunity in CAR T cells.

**Abstract:**

CAR T cell treatment of solid tumours is limited by poor persistence partly due to CD95 ligand (CD95L)-induced apoptosis. Both T cells and cells within the tumour microenvironment (TME) may express CD95L, triggering apoptosis in CD95-receptor-positive CAR T cells. Tonic signalling of CAR T cells may also increase CD95-dependent AICD. Because the intracellular protein c-FLIP protects T cells from AICD, we expressed c-FLIPp43 within a Her-2 targeted CAR cassette and evaluated the potential of c-FLIPp43 through in vitro functional assays and in vivo tumour-bearing xenograft model. cFLIP expression protected against CD95L-induced cell death in the Jurkat T cell lines. However, in primary human CAR T cells containing CAR-CD28 domains, c-FLIPp43 overexpression had minimal additional impact on resistance to CD95L-induded cell death. In vitro cytotoxicity against a breast cancer tumour cell line was not altered by c-FLIPp43 expression, but the expression of c-FLIPp43 in Her2-CAR T cells lowered interferon-γ secretion, without markedly affecting IL-2 levels, and c-FLIPp43-Her2-CAR T cells showed reduced anti-tumour activity in immunodeficient mice with breast cancer. The findings of this study provide a new understanding of the effects of controlling extrinsic apoptosis pathway suppression in CAR T cells, suggesting that c-FLIPp43 expression reduces anti-tumour immunity through the modulation of effector T cell pathways.

## 1. Introduction

To date, the FDA has approved six CAR T cell therapies to treat blood cancers, with products including Abecma, Breyanzi, Kymriah, Tecartus, Yescarta and Carvykti [1,2,3,4,5,6]. Despite the promising outcome in treating B cell malignancies, CAR T cell treatment in solid tumours has yet to achieve complete remission [7,8]. Solid-tumour-directed CAR T cells face multifactorial challenges because of the immunosuppressive TME [9], heterogeneity of tumour-associated antigens (TAAs) [10,11], on-target and off-tumour toxicity [12,13,14], inefficient CAR T cell trafficking [15] and the upregulation of inhibitory receptor and death ligands [16,17,18]. The components of the immunosuppressive TME such as tumour cells, regulatory T cells and myeloid-derived suppressor cells (MDSCs) secrete CD95 ligand (CD95L), inducing CAR T cell apoptosis, resulting in short-term persistence in vivo [19,20,21,22]. Adoptively transferred CAR T cells also express and secrete high levels of CD95L, resulting in further enhancement of activation-induced cell death (AICD), compounded by ex vivo expansion and recursive in vivo antigen activation [23,24,25]. The binding of CD95 to a CD95L receptor triggers the extrinsic apoptosis pathway through the formation of death-inducing signalling complex (DISC) and initiates the caspase-dependent apoptosis pathway in CAR T cells [25,26]. Several studies have demonstrated that the performance of CAR T cells for solid tumours could be enhanced through the disruption of the CD95 signalling pathway [27]. Various strategies such as (1) generation of dominant-negative CD95 receptor [17], (2) integration of Bcl-2 molecules [28], Mcl-1 [29] and (3) Bcl2L1 [30] have been utilised to improve the efficacy of CAR T cell therapy in solid and blood tumours. c-FLIP is a well-established anti-apoptotic gene that inhibits the extrinsic apoptosis pathway, including CD95, tumour necrosis factor-α (TNF-α) and the TNF-related apoptosis-inducing ligand (TRAIL)-dependent apoptosis pathway [31,32,33]. c-FLIP protects mature T cells from T cell receptor (TCR)-induced apoptosis [34], suggesting that co-expression of the c-FLIP transgene has potential to promote CAR T cell survival. However, there are no reported studies that have investigated the functional potential of c-FLIP expression in CAR T cells for treatment of haematological or solid malignancies. Therefore, this study aimed to integrate the anti-apoptotic function of c-FLIPp43 in CAR T cells to overcome AICD against solid tumours. However, the expression of c-FLIPp43 in Her2-CAR T cells reduced protection against human Her-2-positive breast cancer in a xenograft model. This study provides further insights on the effects of modulating the CD95/CD95L pathway and the possible additional roles of c-FLIPp43 transgene in T cell cytolytic pathways.

## 2. Materials and Methods

### 2.1. Plasmid Construction

The third-generation lentiviral pCCLsin.cPPT.hPGK.GFP.WPRRE (pCCLsin) transfer vector, pMD2 (VSVg), pMDLgpRRE (gag-pol) and pRSV-Rev (rev) packaging vectors were a kind gift from Prof. Luigi Naldini [35]. The anti-Her2 CAR FRP5 has been previously described [29,36]. The c-FLIPp43 gene was codon-optimised and synthesized by IDT Technologies. Green fluorescent protein (eGFP) was subcloned from pSBtet-GP (Addgene #60495). Both CAR constructs are second-generation CAR with CD28 co-stimulatory domains and consist of a EQKLISEEDL c-myc tag between scFv and CD8 hinge [37]. GFP and c-FLIPp43 were integrated into CAR cassette, separated from the CAR using self-cleaving 2A (P2A) peptide sequences. The scarless assembly of Her2CAR-P2A-GFP-P2A-cFLIPp43 (Her2CAR-cFLIPp43) was performed using SapI Type IIs restriction enzyme and was further cloned into pCCLsin into the BamHI and SalI restriction sites. The construction of pcDNA3.1(-) LZ-CD95L (Addgene #104349) has been previously described [29].

### 2.2. Cell Culture

Cell lines including HEK293T (ATCC CRL-1573) and MCF-7 (ATCC HTB-22) were cultured in high-glucose Dulbecco’s Modified Essential Medium (DMEM) supplemented with 10% fetal bovine serum (Pan-Biotech GmBH, Aidenbach, Germany)), penicillin (100 U/mL) and streptomycin (100 µg/mL) (Gibco) in a humidified incubator at 37 °C, 5% CO_2_. MCF-7 and HEK293T cells were transfected using Lipofectamine 3000 according to manufacturer’s protocol. Human peripheral blood mononuclear cells (PBMC) were derived from healthy donors with written consent and approval from the University of Otago Human Ethics Committee (Health; Ethics Approval # H18/089). PBMCs were isolated using Ficoll-Paque density-gradient centrifugation, followed by T cell isolation using an EasySep Human T cell isolation kit (Stemcell Technologies, Vancouver, BC, Canada). Isolated T cells were cultured in T cell expansion media (Thermo fisher, Waltham, MA, USA, #A1048501) supplemented with 50 U/mL of hIL-2 (Peprotech, East Windsor, NJ, USA, #200-02), L-glutamine and 10 U/mL penicillin and streptomycin (Gibco) and activated with Dynabeads Human T-activator CD3/CD28 at 1:1 ratio for 48 h prior to transduction.

### 2.3. Lentiviral Particle Production

Virions were packaged in HEK293T cells using Lipofectamine 3000 transfection method according to manufacturer’s protocol (Thermofisher). Briefly, 2.2 × 10^7^ cells were plated in T_175_ flasks in DMEM medium supplemented with 10% fetal bovine serum one day prior to packaging. Cells were then transfected with 41.4 μg of packaging plasmids at 1:1:1 ratio (pRSV-Rev, pMDLg/pRRE, pCMV-VSV-G) and 13.8 µg transfer plasmid (pCCLsin). The next morning, the medium was gently replaced with 32 mL of pre-warmed OptiMEM containing 5% fetal bovine serum, penicillin and streptomycin for a further 48 h incubation at 37 °C, 5% CO_2_. Medium was then harvested, followed by ultracentrifugation at 120,000× *g* for 2.5 h at 4 °C, and stored at −80 °C until use. To determine titres, HEK293T cells were transduced with lentivirus at different dilutions in the presence of 8 μg/mL of polybrene (Sigma-Aldrich) as previously described [29].

### 2.4. Generation of CAR T Cells

One day prior to transduction, plates were coated with 40 µg/mL retronectin (Takara Bio, San Jose, CA, USA, #T100A/B) at 4 °C, blocked with 2% FBS/PBS, followed by spinoculation at 800× *g* for 2.5 h at room temperature. Activated T cells were then transduced with lentivirus encoded with CAR constructs at a multiplicity of infection (MOI) of 40. The next day, T cells were debeaded and expanded in media supplemented with 50 U/mL of hIL-2. Media was changed with fresh medium every three days, maintaining T cell concentration at 1 × 10^6^ cells/mL for 14 days for functional analyses.

### 2.5. RNA Extraction, cDNA Synthesis and qPCR

RNA of transduced T cells was extracted using NucleoSpin RNA Plus kit (Macherey-Nagel, Duren, Germany) and reverse transcribed using PrimeScript™ RT Reagent Kit (Takara Bio, USA) according to manufacturer’s protocol. qPCR was performed using primers c-FLIPp43 Fwd: CGCAGAGTATCCCAGAGGAG and c-FLIPp43 Rev: CCCCTTGACACCAGAACG. Housekeeping gene β-actin was used as an internal control to normalise the target gene expression with β-actin Fwd: CTTCCTTCCTGGGCATG and β-actin-Rev: GTCTTTGCGGATGTCCAC.

### 2.6. Immunoblot Analysis

Cell lysates were prepared using RIPA lysis buffer and blotting was carried out using rabbit anti-human c-FLIP (Abcam, Cambridge, UK, #ab28421), and mouse monoclonal β-actin primary antibody (Sigma-Aldrich , St. Louis, MO, USA, #A2228) was used as loading control. Goat anti-mouse IgG DyLight 680 (Thermo Fisher, #A3274) and goat anti-rabbit IgG 800 at 1:10,000 dilution were used as secondary antibodies. The membrane was scanned using an Odyssey Fc imaging system (Licor, Bad Homburg, Germany) and analysed using Image Studio Lite software.

### 2.7. Mitochondrial Membrane Potential Assay (TMRE)

On day 7 post-transduction, transduced primary T cells were incubated overnight with 1 μg/mL LZ-CD95L, and then 4 μM TMRE (Invitrogen, Waltham, MA, USA) was added at 37 °C for 30 min. DAPI (50 ng/mL) was added immediately prior to flow cytometric analysis and GFP positive cells were electronically gated for quantification of TMRE and DAPI signals using the YG586/16 and BV421 channels respectively.

### 2.8. Cytotoxicity and Cytokine Release Assay

The luciferase-based cytotoxicity assay was carried out as previously described [38]. On day 7 post-transduction, effector CAR T cells were co-cultured with target MCF-7 (Her2^+^) tumour cells at a 10:1 ratio of effector to target (E: T) cells in triplicate using Firefly Luc One-Step Glow assay (Thermo Fisher #16197). To evaluate the role of TNF-α in CAR T cells killing breast cancer cell line, 5.0 × 10^6^ of each CAR T cell constructs were treated with 100 μg/mL infliximab (IFX) and incubated for 24 h at 37 °C, 5% CO_2_ prior to cytotoxicity assay. For IL-2 and IFN-γ cytokine release assay, CAR T cells were stimulated with Her2^+^ target cells at 2:1 ratio. The concentration of cytokines secreted in cell supernatant was measured 24 h post-incubation using sandwich ELISA according to manufacturer’s protocol (BD Biosciences, San Jose, CA, USA). Plates were read on a Varioskan Lux multimode microplate reader (Thermo Fisher, USA). To measure intracellular cytokines, perforin and granzyme, CAR T cells were co-incubated with target cells at 2:1 ratio for 6 h and brefeldin A for 4 h (Biolegend, San Diego, CA, USA, #420601) prior to intracellular labelling.

### 2.9. Flow Cytometry Analysis

Cells were subjected to forward and side scatter for doublet discrimination, and dead cells were excluded from analysis using Zombie NIR viability dye (Biolegend #423106). For cell surface staining, cells were incubated in FACS wash buffer (0.5% BSA/2mM EDTA/PBS) with antibodies for 15 min followed by incubation with streptavidin-conjugated APC. Cells were then fixed with fixation buffer (0.1% paraformaldehyde/0.1% BSA/PBS) for 15 min at room temperature. For intracellular staining, cells were permeabilized with permeabilization buffer (0.2% Tween-20/1% BSA/0.02% azide/PBS) for 30 min at 4 °C and further incubated with antibodies recognising perforin and granzyme B diluted in permeabilization buffer for 30 min at 4 °C. The fluorescent labelled monoclonal antibodies used in this study were: anti-CD3 antibody (Biolegend, #300424), biotin anti-c-myc antibody (Biolegend, #908805) detected with Strepavidin-Brilliant Violet 421 (Biolegend, #405225), anti-CD69 (Biolegend, #31090), anti-perforin (BD, #563673) and anti-granzyme B (Invitrogen, #MHGB05). Flow cytometric data were acquired using a BD LSRFortessa with BD FACSDiva software. Data were analysed with FlowJo v10.6.2 software.

### 2.10. In Vivo Anti-Tumour Activity of CAR T Cells

Age-matched female NOD-scid IL2Rg^ammanull^ (NSG) mice (*n* = 8 per treatment) were purchased from Jackson laboratories (Sacramento, CA, USA) and bred and maintained in individual ventilated cages under specific pathogen-free conditions, in accordance with the protocol and guidelines approved by the University of Otago Animal Ethics Committee. Mice were subcutaneously injected in the flank with 2 × 10^6^ Her-2^+^ antigen-expressing MCF-7 cells. Ten days later, mice were randomly assigned to groups and treated with PBS, 1 × 10^7^ pCCLsin-transduced T cells, Her2-CAR T cells or Her2-CAR-cFLIPp43 CAR T cells isolated from two healthy donors. The tumour burden of the mice was measured with digital callipers every second day using the formula: V = 0.5 × length × width^2^. Mice were sacrificed once the tumour burden exceeded 1000 mm^3^.

### 2.11. Statistical Analysis

All experiments were presented as the mean ± standard error of mean (SEM) and analysed by Student’s *t*-test, two-tailed paired *t*-test, and one-way or two-way ANOVA test with Bonferroni post-test correction. The *p* values of ≤ 0.05 were considered statistically significant (* *p* < 0.05, ** *p* < 0.01, *** *p* < 0.001, **** *p* < 0.0001).

## 3. Results

### 3.1. Expression of Transgene c-FLIPp43 in Primary T Cells

Transgenes comprising anti-Her2-CAR, GFP and c-FLIPp43 were cloned into a multicistronic cassette using a self-cleaving 2A (P2A) peptide strategy (Figure 1A). c-FLIPp43 was codon-optimised for the purpose of detecting the endogenous gene expression using c-FLIPp43-specific primers and to enhance translation. qPCR and immunoblot analyses were carried out on primary T cells or HEK293T cells to confirm the ability of the cassettes to drive transgene expression. CAR T cells displayed a significant mRNA expression of c-FLIPp43 and Her2CAR-cFLIPp43 (29.3-fold expression ± 1.84 SEM), as compared to control Her2-CAR T cells (Figure 1B). Protein expression of transgene c-FLIPp43 was confirmed via immunoblot analysis with c-FLIP antibody. Bands at 43-kDa were detected in Her2CAR-cFLIPp43 CAR T cells (relative induction of 50.9 ± 16.3 SEM), indicating exogenous expression of c-FLIPp43 (Figure 1C).

### 3.2. Functional Effects of c-FLIPp43 Expression in Her2-CAR T Cells

To investigate the functional and phenotypic effects of c-FLIPp43 expression in CAR T cells, primary human T cells were isolated, CD3/CD28-activated and transduced with lentiviral constructs (pCCLsin, Her2-CAR and Her2CAR-cFLIPp43). Primary human T cells transduced with lentiviral constructs were gated based on live cells, CD3^+^ and GFP^+^ expression prior to further intensity analysis. First, CAR surface expression between the Her2-CAR and Her2CAR-cFLIPp43 CAR T cells was compared by flow cytometry analysis using a c-myc tag antibody (Figure 2A). The c-myc tag antibody detects a minimal c-myc peptide sequence located in the hinge region of all Her2-CAR cassettes. The dual expression of CAR and cFLIP in Her2CAR-cFLIPp43 CAR T cells resulted in a non-significant decrease in CAR expression (MFI 774 ± 71 SEM), as compared to Her2-CAR T cells (MFI 1963 ± 546 SEM). Next, the cytotoxicity and activation of CAR T cells were evaluated following incubation of CAR T cells with target cells of Her2^+^ Luciferase^+^ MCF-7 breast cancer cell line. Although the CAR expression of Her2CAR-cFLIPp43 CAR T cells was reduced, the two CAR T cells displayed similar killing at both 24 h and 48 h post-incubation (Figure 2B). The expression of CD69^+^ marker as an early activation marker also displayed a non-significant decrease in Her2CAR-cFLIPp43 CAR T cells (79.7% ± 16.2 SEM) compared to Her2-CAR T cells (84.8% ± 11.4 SEM), possibly due to reduced CAR expression in Her2CAR-cFLIPp43 CAR T cells (Figure 2C).

### 3.3. c-FLIP Expression Protects CAR T Cells from CD95L-Induced Apoptosis

To evaluate the anti-apoptotic activities of c-FLIP, we initially tested Jurkat cells that were stably transposed with Tet-On Sleeping Beauty cassettes containing cFLIP isoforms (cFLIP and cFLIP-p43) as the only genes of interest, without additional CAR sequences. As shown in Appendix A, overexpression of cFLIP in Jurkat was able to protect against CD95L-induced mitochondrial depolarisation and cell death (Appendix A). In the absence of doxycycline induction, protection against CD95L triggering was minimal, showing that the protection was linked to the onset of cFLIP gene expression. Next, primary human Her2-CAR T cells (with or without cFLIP-p43) were challenged with 1 μg/mL LZ-CD95L to mimic AICD. In the absence of LZ-CD95L, no difference in T cell viability was observed between CAR T cell types (Figure 3). T cells transduced with pCCLsin control showed the greatest susceptibility to LZ-CD95L challenge. As the Her2-CAR construct consisted of a CD28 costimulatory domain, Her2-CAR T cells likely displayed an inherent and intermediate resistance against LZ-CD95L [29], and no additional effect of cFLIP expression was noted in protection against LZ-CD95L challenge (Figure 3).

### 3.4. Effect of c-FLIPp43 Expression on Antigen-Triggered Cytokine Release

c-FLIP-L and p43 isoform expression has been reported to enhance T cell proliferation [39] and NFkB activation [40]. We further evaluated the functional effects of c-FLIPp43 overexpression in CAR T cells through cytokine secretion and T cell cytolytic pathways (Figure 4). Her2CAR-cFLIPp43 CAR T cells showed a slight but non-significant increase in IL-2 (37.7 ng/mL ± 11.21 SEM) over three donors compared to Her2-CAR T cells (27.5 ng/mL ± 8.78 SEM). However, overexpression of c-FLIPp43 in Her2-CAR T cells appeared to diminish IFN-γ cytokine secretion (38.8 ng/mL ± 1.14 SEM) compared to Her2-CAR T cells (8.8 ng/mL ± 2.56 SEM). Since Her2CAR-cFLIPp43 CAR T cells exhibited reduced IFN-γ, additional cytolytic effector mechanisms (perforin/granzyme pathway) were investigated [41,42]. However, the expression of c-FLIPp43 in Her2-CAR T cells did not significantly reduce perforin or granzyme expression (Appendix A).

### 3.5. c-FLIP Expression Reduces In Vivo Anti-Tumour Activity in Xenograft Tumour Mice Model

To further investigate the impact of c-FLIPp43 overexpression on the anti-tumour activity of CAR T cells, NSG mice were implanted with a breast cancer cell line expressing the Her2 antigen. Mice were monitored for 90 days, and the survival for each group was plotted (Figure 5). Mice from the non-treated (PBS) group and pCCLsin control treatment group reached the experimental humane endpoint first, with a median survival of 35 days (Figure 5B). Her2CAR-cFLIPp43 CAR T cells (compared to Her2-CAR T cells) lowered the survival of tumour-bearing mice (Figure 5, *p* < 0.001). Although mice treated with Her2CAR-cFLIPp43 CAR T cells displayed tumour reduction, the tumour relapsed during the later days of observation. Only two out of eight mice treated with Her2CAR-cFLIPp43 CAR T cells eliminated their tumour burden, while six reached the humane end point. In this study, 2 out of 40 mice (one each from Her2-CAR T cells treatment group and Her2CAR-cFLIPp43 CAR T cells treatment group) developed GvHD-like symptoms after day 40 post-tumour-challenge.

## 4. Discussion

Several studies have sought to attenuate the extrinsic apoptosis pathway in CAR T cells, with the goal of improving persistence and anti-tumour activity [17,27,28,43]. To date, this is the first study to investigate the functional roles of c-FLIP expression in CAR T cells in solid tumour treatment. In this study, lentiviral cassettes expressing Her2-CAR and Her2CAR-cFLIPp43 were utilised for functional testing in primary T cells. The cassette produced high lentiviral titres and high transduction efficiency and displayed significant induction levels of transgenes (Her2-CAR, GFP and c-FLIPp43) in primary T cells, allowing functional in vitro and in vivo studies. Previous studies have shown that c-FLIP isoforms are involved in anti-apoptotic regulation of the extrinsic apoptosis pathway [31]. Expression of c-FLIP isoforms can be induced by TCR activation to attenuate CD95-induced apoptosis [44]. However, these studies were performed in unmodified T cells and not CAR T cells that already contain anti-apoptotic CD28 domains. We and others have previously noted a significant contribution of tonic CAR signalling to protection from AICD, most likely mediated by the CAR CD28 domain, with survival effects apparent even in the absence of antigen triggering [29,45,46].

Co-expression of multiple genes using polycistronic vector often results in decreased protein expression, as transcript length of the transgenes may inversely correlate with the protein expression [47,48]. In this study, Her2CAR-cFLIPp43 CAR T cells displayed a small but non-significant decrease in CAR expression compared to the Her2CAR T cells, further highlighting the utility of the P2A system for multigene expression in CAR T cells. Previous overexpression studies of c-FLIP in T cells displayed conflicting results concerning T cell activation. In 2000, Kataoka et al. reported that c-FLIP_L_ induces T cell proliferation through the activation of NF-κB and ERK pathways [49]. However, in 2004, Tai et al. revealed that the thymocytes of c-FLIP-L-transgenic mice displayed a lower frequency of CD69^+^ expression and reduced IL-2 production upon CD3 stimulation [50]. Ohme et al. 2005 reported similar findings, suggesting that the overexpression of c-FLIPs isoform impairs the proliferation of T cells in transgenic mice [51]. However, the canonical activation in T cells requires the full machinery of TCR-MHC recruitment (CD3εγ and CD3ζ), while CAR T cell activation only requires CD3ζ domain [52]. Therefore, observation in T cells and the expression of c-FLIP isoforms in general may not be applicable to the findings of this study. In this study, Her2-CAR-c-FLIPp43 CAR T cells exhibited similar frequency of CD69^+^ expression (compared to Her2-CAR T cells) upon antigen stimulation, suggesting the overexpression of c-FLIPp43 in CAR T cells has no observable impact on this T cell activation marker.

Dohrman et. al. found in 2005 that CD8^+^ T cells overexpressing c-FLIP_L_ produced higher levels of IL-2 with enhanced T cell proliferation [53]. Although IL-2 release of Her2CAR-cFLIPp43 CAR T cells was overall slightly higher over three donors, this effect failed to reach statistical significance. c-FLIPp43 protein has been shown to bind with RIP1 protein leading to the activation of NF-κB pathway [49]. It is possible that the overexpression of c-FLIPp43 enhanced RIP1-interactions, inducing IL-2 release through the activation of NF-κB pathway in CAR T cells. However, this hypothesis has yet to be proven using CAR T functional assays. IFN-γ is a pro-inflammatory cytokine secreted upon CAR T cell activation to promote the differentiation and anti-tumour potential of cytotoxic T cells [54,55,56]. IFN-γ also enhances T cell infiltration at the tumour site through the induction of chemokines (CXCL9, CXCL10 and CXCL11) [57,58]. Kylaniemi et al. revealed that the overexpression of c-FLIP_L_ differentiated CD4^+^ T cells into the TH_2_ subset, resulting in a reduction in IFN-γ cytokine production upon TCR stimulation [59]. This finding is in line with this study’s outcome, as Her2CAR-cFLIPp43 CAR T cells secreted a significant lower level of IFN-γ upon co-incubation with Her2^+^ target cells. It is also possible that the ectopic c-FLIPp43 expression led to epigenetic alterations in CAR T cells, resulting in differentiation into a subject compromised for IFN-γ secretion. Cytotoxic T lymphocytes utilise several overlapping or redundant pathways for targeted killing that include perforin/granzyme-mediated, CD95L and TNF-α apoptosis [60,61].

Overexpression of c-FLIPp43 had no influence on the tumour lytic activity of CAR T cells. However, despite exhibiting similar cytocidal activity, a reduction in perforin and granzyme B activity was observed in two of three donors expressing Her2CAR-cFLIPp43 CAR T cells. The findings open the possibility that Her2CAR-cFLIPp43 CAR T cells utilise other calcium-independent pathways such as CD95/CD95L or TNF-α as the major killing axis [60,62,63,64]. However, CAR T cells treated with anti-TNFα were not impaired in tumour killing relative to the cFLIP expressing CAR T cells. This study did not confirm that perforin and granzyme release was altered in antigen-triggered cFLIPp43 CAR T cells. Therefore, a loss of granzyme and perforin secretion is unlikely to explain the poor in vivo anti-tumour activity of cFLIPp43 CAR T cells.

Alderson et al. demonstrated that, in the presence of CD3 mAb, an anti-CD95 mAb could co-stimulate proliferation and IFN-γ and TNF-α secretion in human T cells in an IL-2 independent manner [65]. It is possible that cFLIP overexpression could inhibit positive CD95 signalling, thereby explaining the loss of T cell effector function observed in our study. Alternatively, the overexpression of c-FLIP in Her2-CAR T cells may have disrupted the levels of caspase production or activation, limiting the cytolytic effector functions of CAR T cells. This speculation is based on Misra et al. 2015, who showed that while caspase-8 activation induces apoptosis, intermediate caspase-8 activity is still necessary for the activation of effector T cells [66]. However, the exact mechanism and extent of caspase activation needed to suppress the T cell effector function has yet to be identified. As c-FLIP has been shown to interact with caspase-8, the most upstream protease of caspase cascade [67,68], the overexpression of c-FLIPp43 in CAR T cells may have altered the balance of pro-enzyme and activated caspases, possibly resulting in suppression of effector T cells.

The in vivo results from the main studies clearly demonstrate that expression of c-FLIPp43 in CAR T cells failed to improve the overall survival in tumour-bearing mice. The results implicate the altered IFN-γ production by Her2CAR-cFLIPp43 CAR T cells in the reduced anti-tumour activity, and this is consistent with previous studies showing a role for IFN-γ-expression in immune responses to solid tumours, without a major requirement for perforin expression [36,69,70]. The role of T cell-derived IFN-γ in the in vivo anti-solid tumour response is likely multifactorial, with IFN-γ effects noted on tumour adhesion molecule and MHC expression, as well as IFN-γ impacting the tumour-associated vasculature [36,69,70].

## 5. Conclusions

This study confirms the role of c-FLIP in promoting cell survival against CD95-induced apoptosis, but c-FLIP expression limited the cytotoxic potential of CAR T cells in vivo. cFLIP overexpression does not appear to be an appropriate method to attenuate CD95-mediated AICD in CAR T cells.

## Figures and Tables

**Figure 1 cancers-14-04854-f001:**
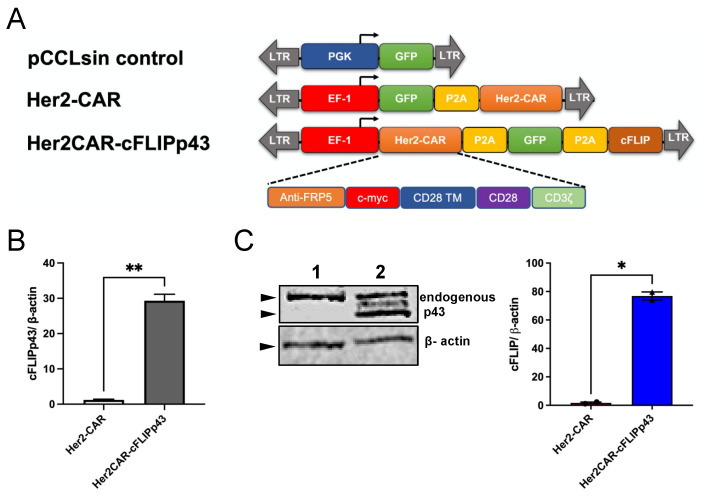
Design of anti-apoptotic c-FLIPp43 CAR T cell cassettes and validation of expression in human T cells. (**A**) Design of lentiviral constructs pCCLsin control, Her2-CAR and Her2CAR-cFLIPp43. pCCLsin control expressing green fluorescent protein (GFP) served as the experimental control for all functional studies. Her2-CAR construct is a c-myc-tagged, second-generation Her2-CAR linked to GFP via porcine teschovirus-1 2A (P2A) peptide. Anti-apoptotic gene c-FLIPp43 was linked to Her2-CAR via P2A peptide to generate Her2CAR-cFLIPp43 construct. LTR, long terminal repeats; PGK, phosphoglycerate kinase promoter; GFP, green fluorescent protein; EF-1, elongation factor-1 promoter; TM, transmembrane domain. (**B**) c-FLIP expression at mRNA level was measured via qPCR analysis. Human primary T cells were transduced with lentiviral constructs at multiplicity of infection (MOI) 40. At 48 h post-transduction, total RNA of transduced T cells was extracted and converted to cDNA. qPCR was carried out using specific primers to analyse the expression of codon-optimised c-FLIPp43. Relative fold change of c-FLIPp43 was normalised against house-keeping gene β-actin. Bar graph shows the mean ± SEM of two different donors. (**C**) Immunoblot analysis to detect protein expression of c-FLIPp43, with β-actin used as a loading control. HEK293T cells (transduced with lentiviral constructs at MOI 2 plus 8 µg/mL polybrene) were lysed with RIPA buffer and processed for immunoblot analysis. Lane 1 was loaded with protein lysate of Her2-CAR construct, whereas lane 2 was loaded with protein of Her2CAR-cFLIPp43 construct. Representative blots are representative data of two independent experiments. Bar graph values represent the quantification of immunoblot analysis (mean ± SEM) using Image Studio Lite. Statistical significance was presented by Student’s *t*-test, * *p* < 0.05 and ** *p* < 0.01.

**Figure 2 cancers-14-04854-f002:**
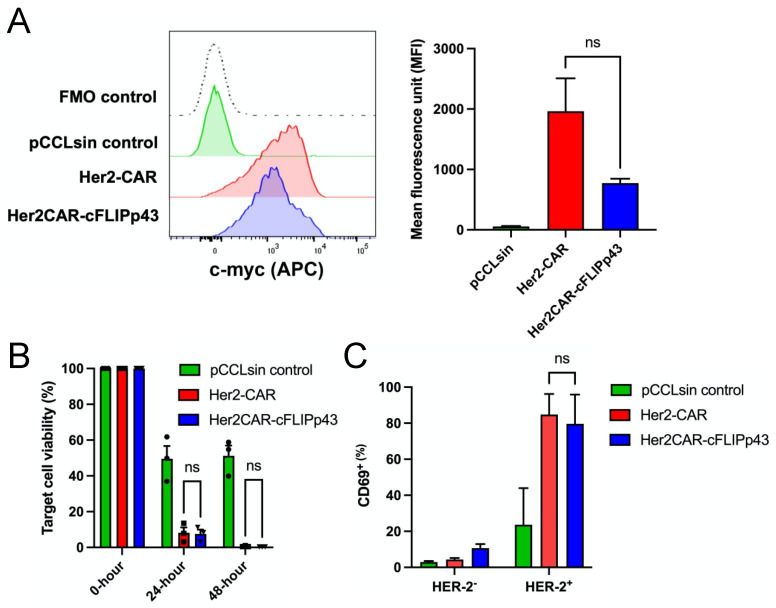
Comparison of CAR expression, in vitro cytotoxicity and CD69 activation marker between Her2-CAR and Her2CAR-cFLIPp43 CAR T cells. (**A**) Representative histogram plots compared the CAR expression (c-myc) between Her2-CAR and Her2CAR-cFLIPp43 CAR T cells at day 7 post-transduction by flow cytometric analysis. T cells transduced with pCCLsin control served as the experimental control. Bar graph displaying the percentage of c-myc^+^ cells (mean ± SEM) from three independent donors. Dead cells were excluded by Zombie NIR viability dye at analysis. Statistical analysis was performed by one-way ANOVA with Tukey’s multiple comparison test (ns: non-significant). (**B**) Luciferase cytotoxicity assay showing specific lysis of Her2-CAR and Her2CAR-cFLIPp43 CAR T cells of Her2^+^ Luciferase^+^-MCF-7 target cells. CAR T cells were incubated with target cells at 10:1 ratio and assessed at 24 and 48 h post-incubation. Bar chart represents the percent of target cell viability (mean ± SEM) of three independent donors, calculated by dividing the luciferase emission of the sample well over the luciferase reading of untreated target cells. Statistical analysis was performed by two-tailed paired *t*-test (ns: non-significant). (**C**) Her2-CAR or Her2CAR-cFLIPp43 CAR T cells were co-cultured with HER-2^+^MCF-7 target cells at 2:1 ratio for 24 h. GFP-positive cells were gated for CD69 expression was then analysed by flow cytometry. Statistical analysis was performed by two-tailed paired *t*-test (ns: non-significant).

**Figure 3 cancers-14-04854-f003:**
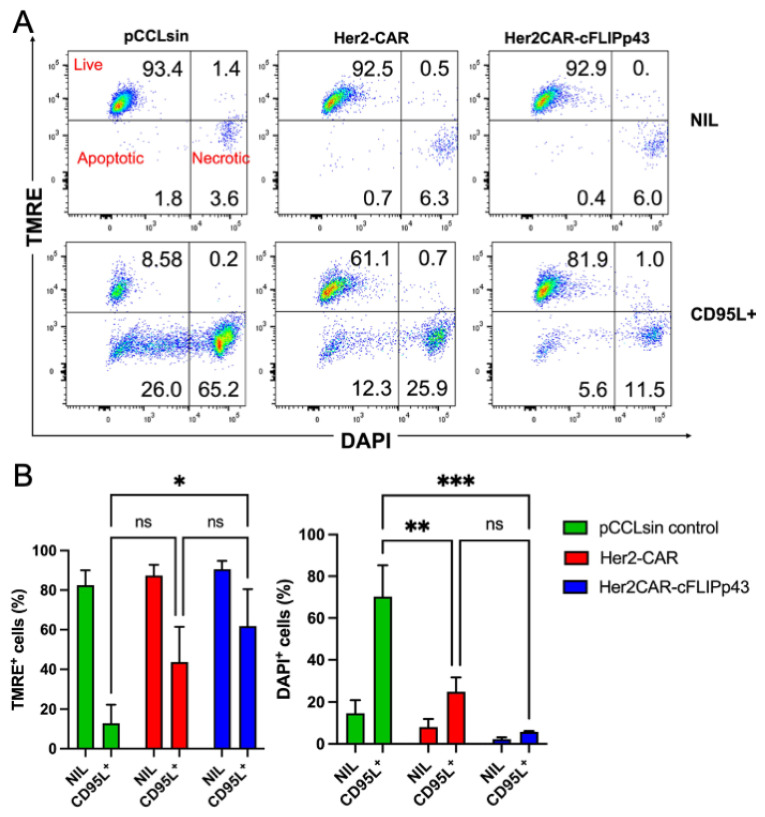
Impact of CD28 domain containing CAR and additional cFLIP-p43 expression on resistance to LZ-CD95L challenge. Primary human T cells transduced with pCCLsin control, Her2-CAR and Her2CAR-cFLIPp43 constructs were challenged with 0 µg/mL (**A**,**B**) or 1 µg/mL LZ-CD95L to mimic activation-induced cell death (AICD). CAR T cells were analysed 24 h post-challenge via TMRE and DAPI staining by flow cytometry. (**A**) Representative dot plots represent the GFP^+^ gated T cells in three quadrants, live cell population (TMRE^+^ DAPI^−^), apoptotic cell population (TMRE^−^ DAPI^−^) and necrotic cell population (TMRE^−^ DAPI^+^). (**B**) Bar graph represents the comparison of live, apoptotic, and necrotic cell population (mean ± SEM) between each of the CAR T cell constructs. T cells transduced with pCCLsin control showed the highest susceptibility towards LZ-CD95L challenge: 8.6% live T cells, 26% apoptotic and 65.2% necrotic. Her2-CAR T cells displayed an inherent and intermediate resistance against LZ-CD95L: 61.1% live T cells, 12.3% apoptotic and 25.9% necrotic. Compared to pCCL-sin T cells, Her2CAR-cFLIPp43 CAR T cells displayed superior protection against LZ-CD95L challenge: 85.9% live, 5.6% apoptotic and 11.5% necrotic. Statistical significance was determined by two-way ANOVA analysis with Tukey’s multiple comparison test, * *p* < 0.05, ** *p* < 0.01, *** *p* < 0.001, ns: non-significant.

**Figure 4 cancers-14-04854-f004:**
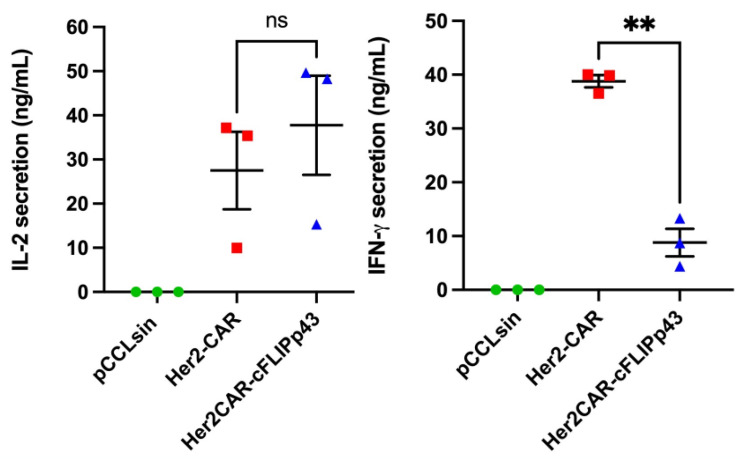
Her2CAR-cFLIPp43 CAR T cells produce similar IL-2 levels as compared to control CAR T cells but exhibit reduced levels of IFN-γ upon Her-2 antigen stimulation. (**Left**) Plots depicting in vitro cytokine concentrations of IL-2 and IFN-γ produced by transduced human primary T cells from three independent donors. At day 7 post-transduction, negative control, pCCLsin-, CAR-control-, Her2-CAR- or Her2CAR-cFLIPp43-transduced T cells were co-incubated with Her2^+^ MCF-7 cell line (E: T, 2:1) for 24 h before supernatant was collected. The cytokine production in the supernatant was detected by sandwich ELISA and was measured in ng/mL. Plots represent mean ± SEM from three independent donors. Statistical analysis was determined by two-tailed paired t test, ns, non-significant; ** *p* < 0.05. (**Right**) Perforin and granzyme B production of CAR T cells groups measured by flow cytometry analysis. Negative control pCCLsin- (green circles), Her2-CAR- (red squares), or Her2CAR-cFLIPp43-transduced T cells (blue triangles) were cocultured at a 2:1 ratio for six hours with Her2-target cells and four hours of Brefeldin A treatment. The cells were fixed and intracellularly stained with perforin and granzyme B antibody for flow cytometric analysis. Each experiment indicates different donor (*n* = 3). Statistical analysis: two-tailed paired *t* test.

**Figure 5 cancers-14-04854-f005:**
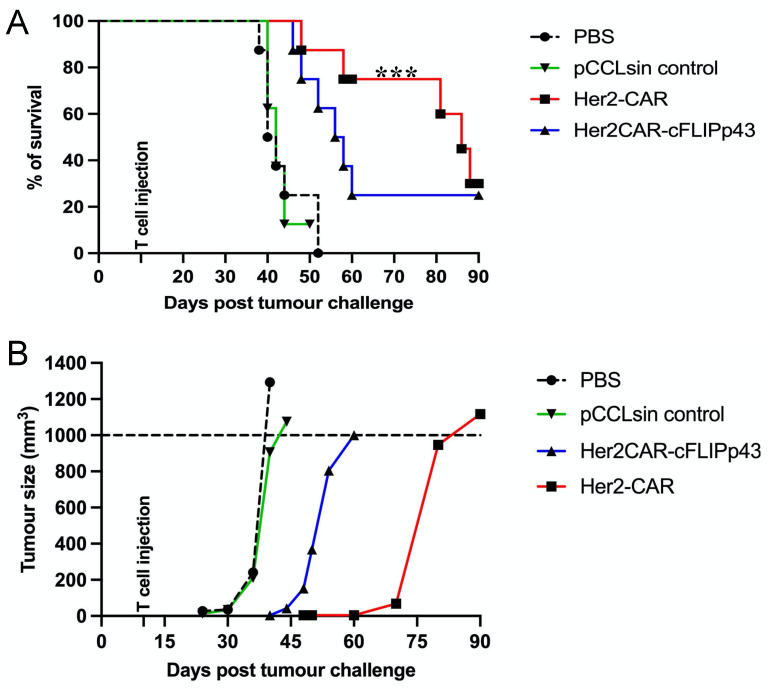
Expression of c-FLIPp43 gene in Her2-CAR reduces anti-tumour activity in vivo. Thirty-two immunodeficient NOD-scid IL2R^gammanull^ mice were subcutaneously injected with 2 × 10^6^ human Her2^+^ MCF-7 cells. The tumour-bearing mice were randomly assigned to groups (*n* = 8) for treatment with PBS, pCCLsin control-transduced T cells, 1 × 10^7^ Her2-CAR T cells or 1 × 10^7^ Her2CAR-cFLIPp43 CAR T cells. Tumour-bearing mice were monitored for 90 days post-inoculation, and tumour volume was measured every alternate day. (**A**) Kaplan–Meier survival curve showing cumulative humane endpoint (survival) differences between treatment groups and (**B**) mean tumour volumes of each group. Statistical analysis using Log-rank (Mantel–Cox) testing, *** *p* < 0.001.

## Data Availability

The data presented in this study are available in this article (and Appendix A).

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
