# Peer review of "Anti-Apoptotic c-FLIP Reduces the Anti-Tumour Activity of Chimeric Antigen Receptor T Cells"

_cancers, 2022, doi:10.3390/cancers14194854_

Round 1

Reviewer 1 Report

The paper by Tan et al. entitled 'Anti-apoptotic c-FLIP reduces the anti-tumour activity of chimeric antigen receptor T cells ' had as objective to integrate the anti-apoptotic function of c-FLIPp43 in CAR T cells to overcome activation-induced cell death against solid tumours. Expression of c-FLIPp43 in Her2-CAR T cells improved T cell survival versus apoptosis and increased IL-2 cytokine secretion upon antigen stimulation compared to Her2-CAR T cells. However, in vitro already a tendency to loss of CAR T cell cytotoxicity was in the case of the expression of c-FLIPp43 in Her2-CAR T cells. This was in agreement with the fact that these c-FLIPp43 expressing CAR T cells   failed to protect from human Her-2 positive breast cancer in a xenograft model.

Major comments

1) This reviewer would prefer that for some in vitro experiments the number of independent repeats is increased since sometimes there are tendencies there but one is not sure if this is a real effect or not. The variability between donors is an important issue so it would be better to increase the n=5 for Figure 4B and figure 5.

2) It is not clear in the results which how the MOI was calculated since the titration of the vectors is completely missing in the method section. The authors should include on which cell line or cells titration was performed and the method (Q-PCR, FACS analysis?)   

3) It would be better to put which MOIS was used in figure 2 for the vectors to be sure that these are valid comparisons.

4) Figure 3 ; Figure 3A and 3B are shown but in the legend A and B is not there

5) Figure 5 the repetition n=? is missing in the legend

6) Better than to put humane endpoint, use experimental endpoint.

7) If P value is not shown in the figure 6 (survival curves) itself while it is mentioned in the legend

Reviewer 2 Report

In this study, the authors showed that CAR-T cells overexpressing c-FLIPp43 exhibited improved anti-apoptotic phenotype but reduced anti-tumour activities in a HER2+ MCF-7 model.

Some of the results are not convincing and need further data support.

Regarding Figure 3, which is the pivotal data to support the anti-apoptotic effect of cFLIPp43, it seems that there's no difference between HER2-CAR-T cells and cFLIPp43-overexpressing CAR-T cells in necrotic or apoptotic cells. The statistical analysis needs to be improved and maybe more technical/biological repeats should be performed. Some of the 'ns' statements are not convincing. and the FACS plot is not representative, e.g. I think the difference in live cell proportion among three types of cells is huge... 81.9% vs 61.1% vs 8.58%. This piece of data is not supportive enough for the conclusion. Additional evidence will be required.

For other results, such as Figure 4 and 5, there are similar issues. I suggest including more biological repeats to achieve statistical significance to support the claim that the overexpression of cFLIPp43 affects the T cell function.

Regarding the in vivo experiment, the authors only showed the survival curve. At least, the efficacy/tumour kinetic should be included. In addition, the authors didn't show any evidence suggesting the anti-apoptotic effect was observed in vivo, e.g. longer in vivo persistence. I also noticed more than 80% similarity between mouse and human CD95L. Not sure the authors have considered the effect of mouse CD95L from monocytes or macrophages.

Overall, I understand that the findings may not satisfy the original hypothesis due to the reduced T cell activity. However, I think if a more detailed mechanism can be revealed, there might be a win-win solution.

Round 2

Reviewer 1 Report

The authors have replied to all my comments and have inserted the requested extra data so

Reviewer 2 Report

Thanks for the revision. I don't have further questions.